# Effectiveness of bimanual coordination tasks performance in improving coordination skills and cognitive functions in elderly

**Danuta Roman-Liu⬭\*, Zofia Mockałło**

Department of Ergonomics, Central Institute for Labour Protection—National Research Institute (CIOP-PIB), Warsaw, Poland

\* daliu@ciop.pl

**Data Availability Statement:** All relevant data are within the paper and its Supporting Information files.

**Funding:** This paper is based on the results of a research task carried out within the scope of the

## Abstract

### Background

The purpose of the study was to determine the impact of the performance of bimanual coordination tasks with specific characteristics on the changes in quality of coordination, musculoskeletal load of the upper limbs and cognitive functions.

### Methods and findings

A group of 26 people aged 60–67 years performed 6 sessions of bimanual coordination training. Each session included set of tasks that varied depending on the shape in which the cursor moved, the coordination mode (in-phase, anti-phase, complex) and the tracking mode (imposed or freely chosen speed). Performance was assessed by: Error, Variability and Execution. The load of upper limb muscles was expressed with the value of the normalized EMG amplitude. Cognitive functions were evaluated using the Vienna Test System. The Variability and Error values obtained during the sixth training session decreased by more than 50% of the initial values. Tasks with freely chosen speed showed changes from 15% to 34% for Error and from 45% to 50% for Variability. For tasks with imposed speed and coordination mode anti-phase or complex it was between 51% and 58% for Error and between 58% and 68% for Variability. Statistically significant differences between load during the sixth training session compared to the first session occurred in three out of four muscles and were between 9% to 39%. There were statistically significant differences in motor time and no differences in variables describing attention and working memory.

### Conclusions

Coordination mode is meaningful for improving coordination skills; tasks in the anti-phase and complex are recommended. Tracking mode also plays a role, tasks with an imposed cursor movement speed have greater potential to improve coordination skills than tasks with freely chosen. Improved control skills resulted in the reduction of upper limb musculoskeletal load. It can be assumed that an increase in coordination skills with the use of appropriate training can help to reduce musculoskeletal load.

fourth stage of the "Improvement of safety and working conditions" National Programme, supported partly in 2017–2019 — within the scope of research and development — by the Ministry of Science and Higher Education/National Centre for Research and Development. The funder had no role in study design, data collection and analysis, decision to publish, or preparation of the manuscript.

**Competing interests:** The authors have declared that no competing interests exist.

## Introduction

As a result of demographic change, the elderly constitute a gradually growing population of workers. This means that not only actions aimed at increasing the quality of life of elderly people, but also at prolonging their professional activity are of great socio-economic significance. The ageing process of the body leads to a gradual degeneration of both the nervous and the muscular system [1]. There are progressive losses in various brain functions, including memory, cognitive functions, or motor control [2]. Reduction of motor functions due to changes in the nervous system and musculoskeletal system can be observed as decreased coordination, increased inaccuracy of movement or slowed movements [3]. This results in the elderly being less accurate when performing a various movement tasks, including coordination tasks [1].

The deficit in the coordination of elderly people, compared to younger subjects, was demonstrated especially in bimanual activities [4,5]. The decline in coordination skills in older people reduces accuracy, smoothness and leads to more errors when they perform tasks requiring bimanual coordination [6,7]. Therefore, elderly people may experience difficulties in carrying out work activities which require comprehensive bimanual coordination [8].

Due to the relationship between the brain and muscle structure and functions, it can be assumed that maintaining the functions of the brain and muscles in good condition is essential for quality of life, including professional life. The functional capacity of elderly workers and/or those with neurological diseases, which is important from the point of view of the work they perform, can be improved with the use of appropriate rehabilitation programes. Therefore, programes that increase muscular strength, overall body performance and proprioception are commonly used [9]. In addition, it is important to improve elderly workers precision in carrying out manipulative activities. The study by Johann et al. [10] showed that coordination training is a more effective tool for improving the quality of tasks requiring coordination than cardiovascular training. The results of the studies on bimanual coordination tasks in rehabilitation confirm the importance of using coordination training to improve the quality of life of elderly workers [11,12].

Research also demonstrates the possible positive impact of bimanual coordination training on brain activity. In particular, bimanual coordination tasks can be represented by a more complex modulation of intra-and inter-hemisphere connections [13], which is important for activating the brain areas associated with cognitive functions. This suggests that training based on bimanual coordination tasks can be a useful tool in mitigating age-related changes, developing motor functions in terms of precision in manual operations, but also in the area of cognitive functions [14,15].

In order to be effective, the rehabilitation programes should produce significant and measurable effects, which means that they should enforce the performance of specific and effective tasks. The measurability of the effects can be ensured by using parameterized tasks on the one hand, and the possibility of measuring error of task performance, on the other. These features provide tasks constructed on the basis of computer technology, when the task is carried out using the force exerted under static conditions on the sensor, and the effects of the force exerted are presented on the screen [16]. Effectiveness of such programs lays also in characteristics of the tasks that are performed. Bimanual coordination tasks are classified in different manners [17,18]. Classification may be based on performance, which may result from: movement of limbs only, movement of limbs with force exertion or from force exerted under static conditions. Tasks can also be differentiated by coordination mode (symmetric and asymmetric), whereas symmetric tasks can be divided into tasks in phase, in anti-phase and complex tasks.

Standardized trainings equipped with adequate measurement variables can both serve rehabilitation purposes and as an indicator of neurodegenerative diseases. Restoring or improving coordination skills after trauma or in the face of chronic consequences of stroke, can be a goal of performance of such training [19,20]. Many devices have been developed to train the upper limbs or to support the training of people after a stroke (for review see vanDelden et al. [20]). The devices differ considerably in terms of their mechanical and electromechanical complexity from relatively simple to complex ones. However, the overwhelming majority of these devices are not adapted to measure the performance. Evaluation of the effects of the rehabilitation training has been done only by means of assessment of the improvement of the post intervention functionality.

Few publications have presented training and their evaluation by quantitative measures. Training sessions were performed once [21,22] or repeated in two, three or five days [15,23–28]. Tasks included in trainings required operating sensors by forearm movements that forced cursor movements on the monitor. Sensors were in a form of handles that slid in parallel along the track from side to side in front of the body [23,27], rotated discs [21,25,26,28] or joysticks [22,24]. The monitor display showed Lissajous figure and tracing lines that had been produced by the participant movements [21,23,27], a blue target line and white dot which follow that line [25,26,28] or two white dots on a black background [24]. In Hoff et al. [15] study participants observed a learning sequence consisting of a 15-letters array on the monitor. Participants pressed the corresponding keys on a keyboard with a corresponding finger when a specific letter occurred. All those studies looked at changes in performance measures in subsequent training sessions. However, the training included only one task and the possible modifications concerned only the number of repetitions and speed of the performance. Given that involvement of brain regions depends on the task characteristics [29] and that task characteristics can affect performance progress in the training of bimanual coordination, there is a need of such a training that would be equipped with various tasks, differentiated according to characteristic features, e.g. trajectory on which the cursor moves, speed of movement, type of the cursor tracking.

The purpose of the study presented in this article was to determine the impact of the performance of bimanual coordination tasks with specific characteristics on the changes in the quality of coordination functions, musculoskeletal loading of the upper limbs and cognitive functions.

## Methodology

### Study subjects

The study covered a group of people aged 60–67, mean 68 (2,9). Body weight and body height were respectively 76 (11) and 180 (8,8). The group included 12 women and 14 men. The subjects were people without neurological, cardiac or musculoskeletal injuries in the past 5 years. The study was approved by Bioethics Commission  of Cardinal Stefan Wyszyński University in Warsaw, approval number: EEiB—12/2017. Prior to the commencement of the study, the study participants were informed about the purpose and course of the study, and signed the consent for participation in 6 bimanual coordination training sessions with various coordination tasks.

### Bimanual coordination tasks stand

The bimanual coordination tasks involved the control of cursors on specified tracks with force. The control was carried out with two stationary joysticks—one per hand. Each joystick was connected to two sensors which measured the torque in the axes perpendicular to each

other. The cursor position on the screen was proportional to the bending moment resulting from the exertion of force by the study subject on the joystick in a given direction. The view of the stand is shown in Fig 1.

The stand consists of the base frame together with guides mounted on the surface of the table and fixed to its worktop; two adjustable, upholstered supports stabilizing the forearm of the subject, mounted on the arms of the base frame; two dual-axle torque meter units with a control holder; and a four-channel tensometric amplifier (model WWU001-4TUSB with USB-AB cable), intended for communication with a computer, with signal sampling frequency selected in software from 20 Hz to 400 Hz per channel and the maximum measurement error <1%. The set also includes a monitor and a software-supplied notebook.

The specialized software package enables: the measurement and 'online' presentation and measurement of parameters of cursor (tracing cursor) position on the monitor screen, which is the end of the vector of the resultant moment of force for both joysticks; registration of values in the X, Y direction processed on measures; presentation of visual stimuli (curve shapes) on the computer screen and in tasks with the imposed speed presentation and measurement of parameters of cursor movement of the target cursor.

Before commencement of the tasks, the measurement of the maximum bending moments in 4 directions (front, rear, left, right) was performed for each hand separately. The initial measurement was used to determine the control force ranges when performing successive tasks, which was 10% of the measured maximum bending moments.

## Bimanual coordination tasks

For the development of tasks for training to enhance coordination functions, the factors which may be responsible for any benefits resulting from such training were considered. Therefore, activities were taken into account which would be effective in the area of improving coordination and improving cognitive functions. The requirements for the tasks included in the training were determined by the characteristics of that task.

The developed training includes continuous tasks related to the exertion of force under static conditions. The value and direction of the force determine the movement of the tracing cursor on the computer screen. The tasks vary depending on the complexity of the task, the shape in which the cursor moves and the tracking mode (freely chosen speed, imposed speed). The window and shape displayed on the screen during the successive tasks subjected to parameterization and quality assessment is shown in Fig 2.

In addition to the curve of specific shape, one or more cursors appeared on the screen. For tasks with imposed speed there were two cursors for each shape, e.g. target cursor and tracing cursor. For tasks with freely choses speed only one cursor (tracking) for one curve shape was present. There were two tasks (FP and FCT) with freely chosen speed (Fig 2). The tasks differed in the shape of the curve and the coordination mode. The FP task was a task in phase, with tracing the shape of two ellipse independently by each hand using an independent cursor. The FCT task consisted of controlling two cursors on curve in a sinusoidal shape. One cursor was moved over a large sine wave with the right hand by means of a right lever and the other cursor over a small sine wave with the left hand.

The remaining tasks were the tasks with imposed speed, i.e. the tasks when the subject was to follow the target cursor moving at the imposed speed with the tracing cursor controlled by the joystick. One of these tasks (IP), was a task in phase, during which the cursors were controlled on tracks resembling a crown. The task was carried out in two steps. In the first step, the movement took place first on the outer perimeter, and later on the inner circumference. In the second step, the cursors started moving from the inner circle. In contrast, the IAP task was

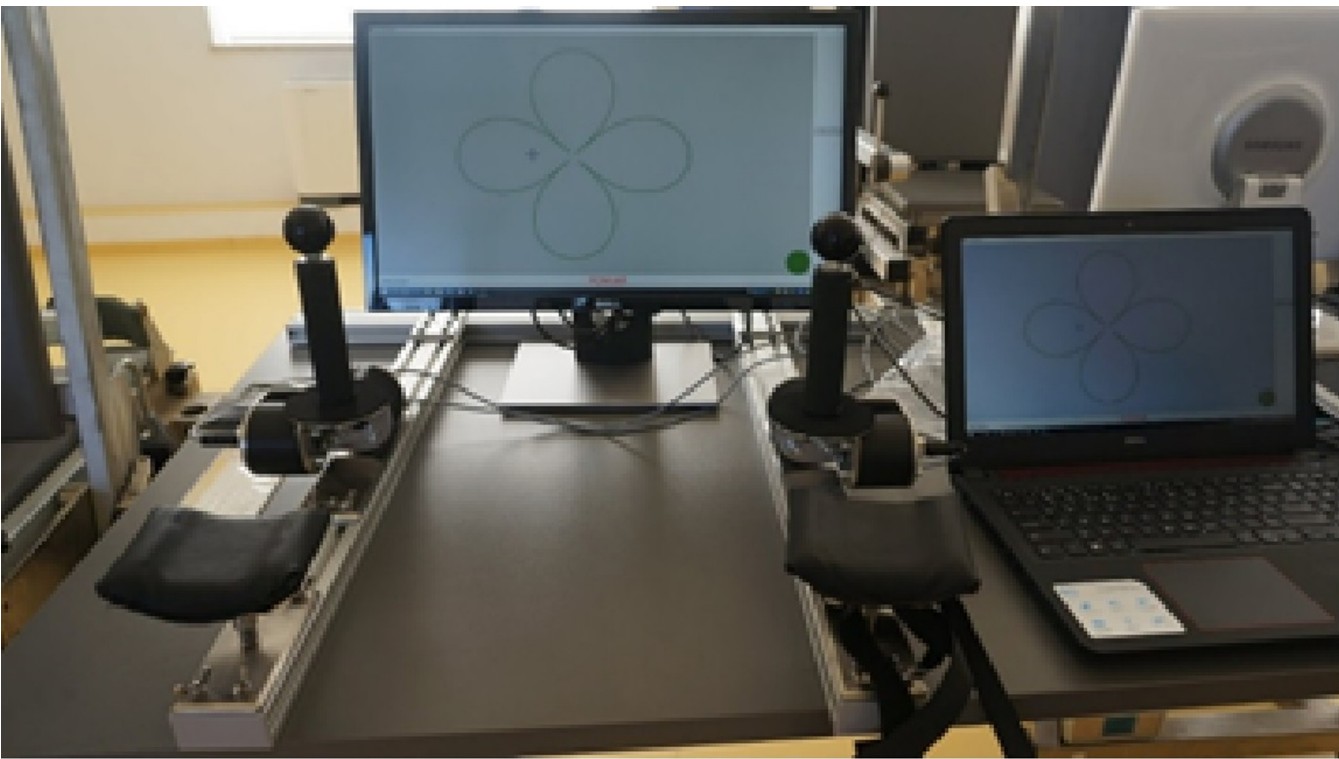

**Fig 1. View of the bimanual coordination tasks stand.**

a task in the anti-phase where the left cursor and the right cursor moved by the study participant were moving on two independent ellipses. The other tasks (ICT, ICOa, ICOb) were complex tasks. In the ICT task, cursors were moving in the opposite direction. In addition, the right and left limbs exerted force on the sensors in mutually perpendicular directions and the velocity in the horizontal direction was twice the speed in the vertical direction. In the ICOa and ICOb task, single-cursor control was carried out using both joysticks, whereas horizontal movement (left-right) corresponded to the left joystick and the vertical movement (up-down) corresponded to the right joystick.

The ability to precisely maintain spatial and temporal requirements is most frequently expressed with accuracy, variability and movement time variables [30]. The error indicators in the studied tasks are calculated on the basis of the difference between the position of the standard curve and the mapping curve corresponding to the tracing cursor. Therefore, the error indicators of each task are calculated as the following measures: Error–the integral value of the difference between the standard curve and the mapping curve, related to the duration of the analyzed task (it determines the reciprocal of precision, that is, the greater the values, the lower the accuracy); Variability–standard deviation of the differences between the standard curve and the mapping curve; Execution–task execution time.

## Study procedure

Participants took part in 6 training sessions training. First session started with an interview with the study participant, a presentation of the purpose of the study and a description of tasks to be performed. After signing the relevant documents, the participant became familiarized

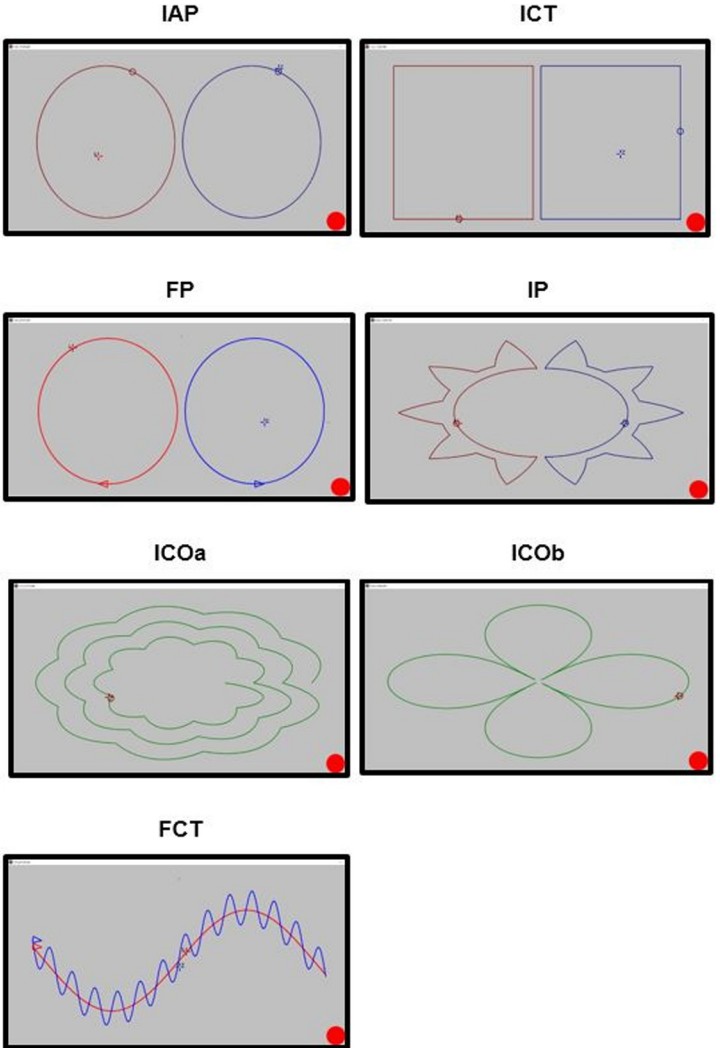

**Fig 2. View of the window appearing during the successive bimanual coordination tasks.** (IAP–imposed anti-phase; ICT–imposed complex; FP–freely chosen in phase; IP–imposed in phase; ICOa–imposed complex; ICOb–imposed complex; FCT–freely chosen complex).

with the bimanual coordination stand and tasks. A study diagram showing the subsequent stages of the study is presented in Fig 3. During each session, the set of bimanual coordination tasks was performed in a single string, that string was performed three times. For the tasks in which the speed of movement of the standard cursor was imposed, the speed was increased by 30% in the second performance and by 50% in the third, in comparison to the first one.

The training consisted of six consecutive sessions at intervals of 2 to 3 days. During the first and the sixth session, to evaluate the effectiveness of bimanual coordination training, tests of coordination and cognitive abilities were conducted using the Vienna Test System. Also, during the first and last session, the EMG signal was registered from four upper limb muscles. Therefore, prior to the first and the sixth session, EMG sensors were placed on the respective muscles on both sides of the body.

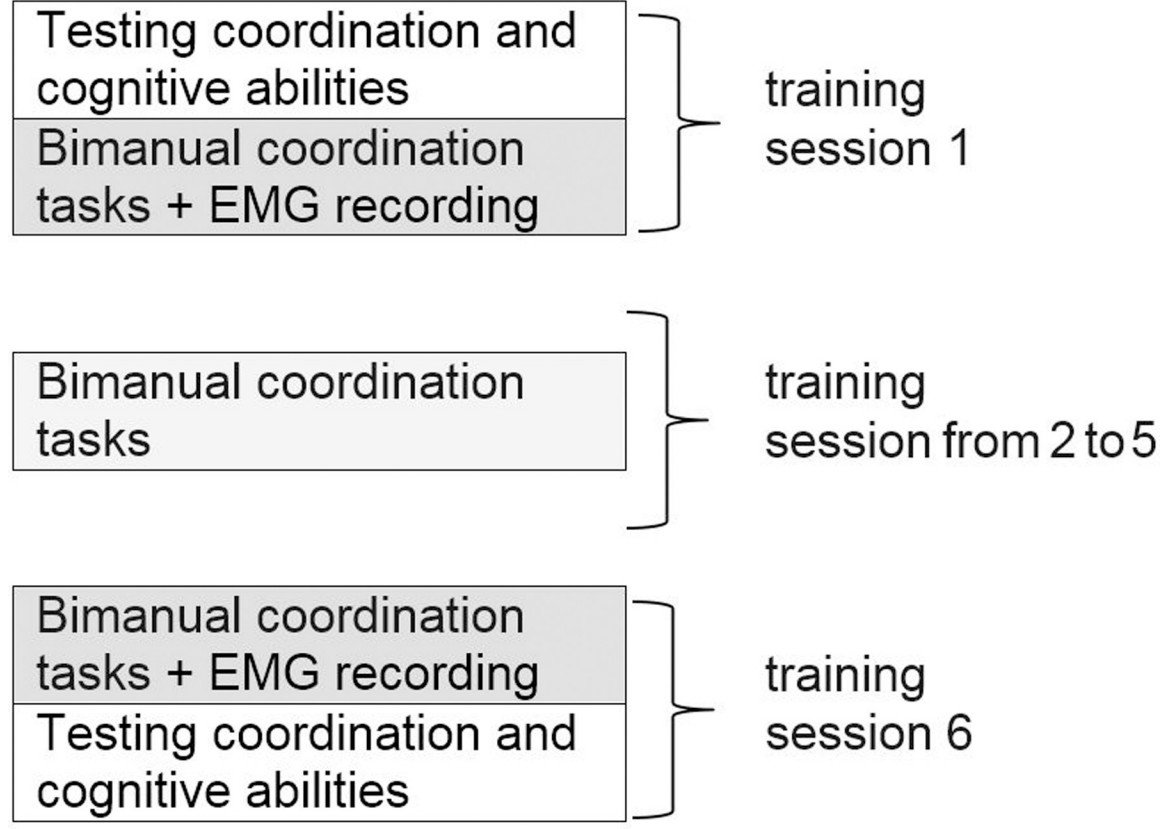

**Fig 3. Diagram showing the successive steps of bimanual coordination training.**

### EMG signal recording and analysis from four upper limb muscles

The performance of manipulative activities with high accuracy increases the level of muscular activity. The increase of functionality in terms of bimanual coordination may result in a reduction of the musculoskeletal loading associated with the performance of activities requiring high coordination and precision. Therefore, the study looked at the load of eight muscles (four muscles of the upper right limb and four muscles of the upper left limb). These were the following muscles: flexor carpi ulnaris (FCU), biceps brachii (BB), lateral deltoid muscle (DE), trapezius (TR).

The EMG signal measurements from those eight muscles were carried out using surface double differential sensors DE-3.1 (Delsys Inc., USA), containing three active electrodes. The electrodes in the sensor were made of silver (99.9% Ag); the single electrode dimensions were 10x1 mm, while the active area of the sensor is 200 mm$^2$ (including both electrode pairs). The input impedance of the sensor is $10^{15}$ Ω, the concurrent signal attenuation coefficient -92 dB, while the initial gain is 10 V/V (for each pair of electrodes).

The placement of measurement sensors on the skin was carried out in accordance with the SENIAM (*Surface ElectroMyoGraphy for the Non-Invasive Assessment of Muscles*). The reference electrode was placed on the back in the area of the lower edge of the scapula. The measuring electrodes for the individual muscles were placed on the skin in the following places:

– FC—on the line from the medial epicondyle of the humeral bone to the pisiform bone at 1/3 of the distance from the medial epicondyle of the humeral bone;

– BB—on the line between the anterior part of the acromion and the olecranon fossa, at 1/3 of the distance from the olecranon fossa;

– DE—at a distance of one finger from the acromion;

– TR—in the middle of the line linking the acromion and the C7 vertebra of the vertebral column.

The EMG signal was recorded with a Bagnoli-16 camera (Delsys Inc., USA). Control of the measuring process was carried out by means of a computer. The EMG signal was recorded with the use of EMG Works 4.0 software (Delsys Inc., USA). The EMG signal sampling frequency was 2 kHz. The transfer band of the apparatus is between 20 and 450 Hz. The signal was amplified with a gain factor of 1,000.

The upper limb load assessment uses the normalized EMG signal amplitude value, calculated as RMS. Normalization of the calculated RMS values, corresponding to the subsequent tests, was based on the RMS value calculated for the test at maximum voluntary contraction of the given muscle, done before the bimanual coordination tests. The RMS parameter is determined from the one-second signal windows (corresponding to 2,000 samples), both for fragments relating to the maximum contractions and for fragments recorded during tests.

## Testing of coordination and cognitive abilities

Coordination abilities were tested as well as cognitive. For this purpose were used tests included in the Vienna Test System (WTS), created and manufactured by the Austrian company Dr. G. Schuhfried Gmbh. In order to assess the level of bimanual coordination of participants prior to and after the 6 sessions of coordination training, two bimanual coordination tests were selected from the test battery: the Two-Hand Coordination Test—2 HAND and the Double Labyrinth Test (DL). The evaluation parameters for coordination in the 2HAND and DL tests were: Total mean duration ($MT_{2HAND}$) (s); Total mean error duration ($ME_{2HAND}$) (s); Total percent error duration ($PE_{2HAND}$) (%); Error duration ($ED_{DL}$) (s); Percentage error duration ($EP_{DL}$) (%).

For the measurement of cognitive functions, attention assessment tests Cognitrone (COG) and the test of information processing Reaction Test (RT) were selected. The following indicators were counted in the COG test: Time of correctly rejected (MTR) (s); Time of correctly accepted (MTA) (s). The following indicators were counted in the RT test: Reaction time [31], which is a measure of reaction speed in response to relevant stimuli (MRT) (ms); Motor time, i.e. measure of the speed of movement in planned action sequences (MMT) (ms). Each indicator was calculated as an average value of series of measurements for each study participant.

## Analysis of the study results

The aim of the study was to determine the impact of six sessions of bimanual coordination training to enhance coordination and cognitive skills. Therefore, an analysis was performed of the differences in the values of the error indicators, the Vienna tests variables obtained before the first session of the training and after the sixth session, and the values of the RMS of upper limb muscles calculated from recordings done during the first session of the training and the during the sixth one. The Wilcoxon Sign Test [32] was used for this purpose. Statistica 10.0 was used for statistical analysis.

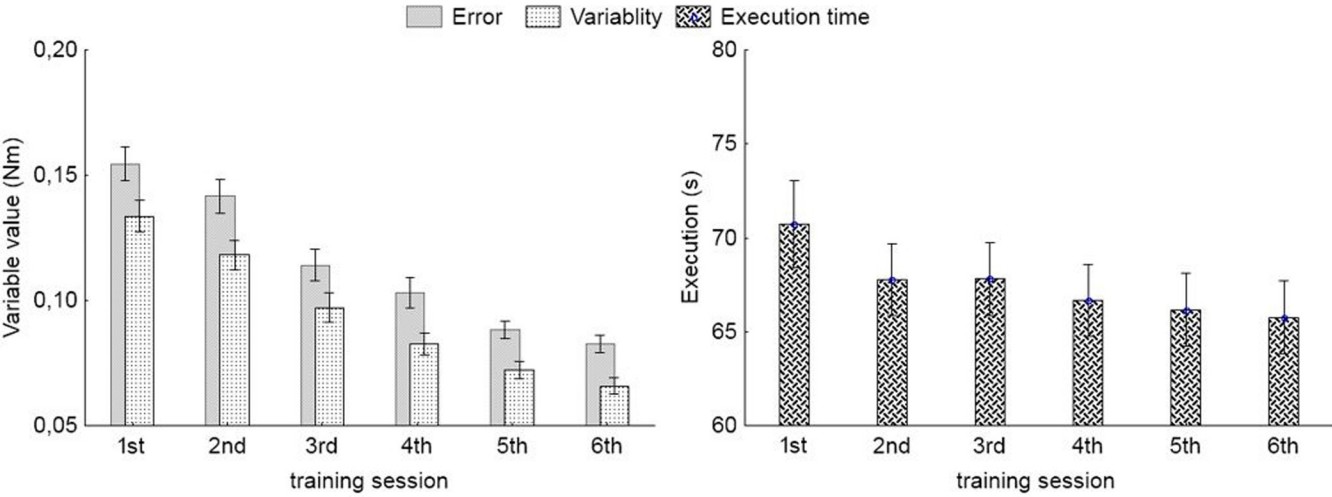

**Fig 4. Mean values and 95% CI of Error, Variability and Execution, obtained during consecutive bimanual coordination training sessions.**

## Results

Fig 4 shows a decrease in the value of error indicators between individual training sessions. The values obtained after the sixth training session decreased by more than 50% of the initial values (first session). The smallest changes occurred with Execution.

Fig 5 presents values showing changes in error ratio, broken down by tasks. The value of the error ratio was calculated as a ratio of values of error indicator obtained during the sixth session to the value of the error indicator obtained during the first session. Values close to 1 denote slight changes; the larger the changes, the smaller the value of the ratio. There are slight differences between the coefficients calculated for Error and those calculated for Variability. The largest values (the smallest changes) occur for the FP, FCT and IP tasks. The smallest values, presenting the largest changes, occurred for the ICOa and IAP tasks, followed by ICOb and ICT.

The results of the Wilcoxon test indicate a statistically significant differences between the values obtained in the first and the sixth training session for all tasks with p<0.001 (Table 1). In case of Error the values of statistic Z were similar for tasks with the imposed cursor movement speed and about 3 times greater than for tasks at freely chosen speed. Also, for the Variability, the tasks with imposed speed had higher values.

Fig 6 shows the normalized amplitude values calculated as RMS for the muscles of the upper limbs. The values obtained from the left and right limb muscles were pooled. Table 2 shows the results of the Wilcoxon Sign Test, presenting differences in the normalized amplitude between the first and the sixth sessions. From among 4 muscles, only the DE muscle lacks the statistically significant differences that would indicate a decrease in musculoskeletal load in the sixth training session compared to the first session. It should be noted that the load level (RMS) of the TR muscle is about twice as high as that of the remaining muscles.

The differences in results obtained by the Vienna Test System during first and sixth sessions are shown in Fig 7 for the tests of coordination and in Fig 8 for test of the attention and information processing. Table 3 shows the results of Wilcoxon Sign Test indicating significance of the differences in the Vienna test measures.

Total mean duration, i.e. the speed of movement ($MT_{2HAND}$) did not change, nor did the number of errors, which means that training sessions did not lead to changes in the speed or

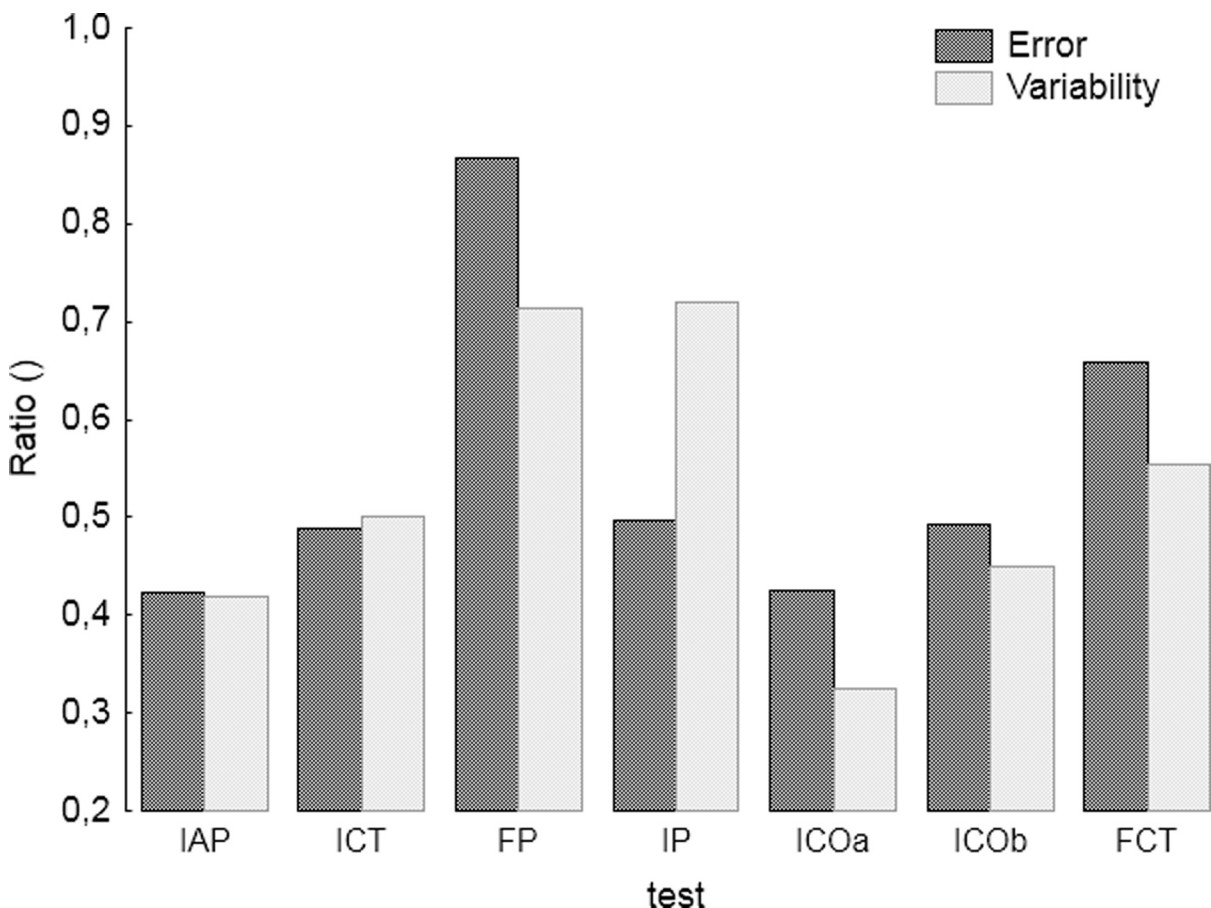

**Fig 5. The value of the coefficient expressing the ratio of Error and Variability calculated during the sixth session to the values obtained during the first session for the seven tasks included in the training.**

quality of coordination tasks. However, the parameter describing the total time where the point was outside the designated route (total mean error duration; ME) and the parameter specifying the ratio of the total time of errors and the total duration of the test (total percent error duration; PE), both in the 2HAND and in the DL test, did change statistically significant.

There were no statistically significant differences in the parameters describing attention. However, statistically significant differences relate to motor time, which is the measure of the speed of movement in planned action sequences, but the direction of this relationship was contrary to what was anticipated. This confirms that the training positively influenced motor functions, but did not affect cognitive abilities.

## Discussion

This article provides an assessment of the impact of bimanual coordination training on the values of variables describing coordination, upper limb muscle load and cognitive performance measures. The bimanual coordination tasks were carried out as force-controlled tasks under isometric conditions. No displacement of upper limbs was an element of standardization of tasks, which favors a standardized assessment of coordination skills [5]. This enabled uniform force on joystick to be accurately recorded.

**Table 1. The results of the Wilcoxon Sign Test showing differences in error indicators between the first and the sixth session of the bimanual coordination training.**

|  | Error | | | Variability | | |
|---|---|---|---|---|---|---|
|  | % | Z | p | % | Z | p |
| IAP | 9.09 | 10.07 | <0.001 | 11.61 | 9.48 | <0.001 |
| ICT | 5.77 | 10.97 | <0.001 | 14.19 | 8.84 | <0.001 |
| FP | 33.77 | 3.95 | <0.001 | 23.87 | 6.43 | <0.001 |
| IP | 8.44 | 10.23 | <0.001 | 12.90 | 9.16 | <0.001 |
| ICOa | 7.79 | 7.29 | <0.001 | 6.41 | 7.59 | <0.001 |
| ICOb | 2.56 | 8.27 | <0.001 | 7.69 | 7.36 | <0.001 |
| FCT | 35.90 | 3.44 | <0.001 | 39.74 | 2.48 | 0.0131 |

Due to the fact that the age-related decline in the ability to learn motor tasks is specific and task-dependent [14, 32], in the tasks developed to improve coordination function, factors have been taken into account which affect the motor function, such as the structure of tasks and their complexity [33]. The tasks developed and used in the studies required continuous exertion of force by bimanual coordination in the directions determined by the shape displayed on the screen and in some cases by the moving target cursor. The selection of continuous tasks was dictated by the results of previous studies demonstrating that the quality of bimanual coordination tasks can be influenced by the decreasing size of the corpus callosum progressing

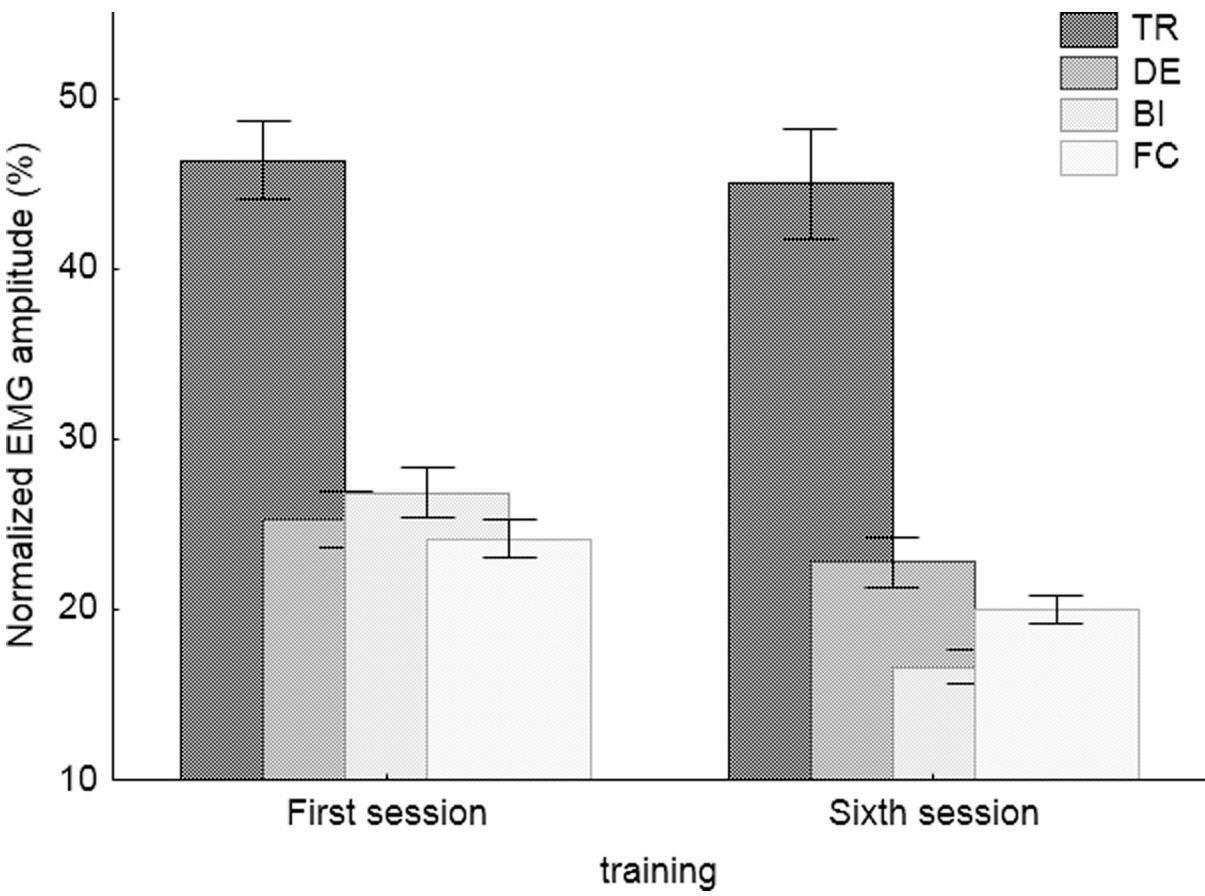

**Fig 6. Mean values and 95% CI of the results of the upper limb muscles activity obtained with electromyography.** (TR—trapezius; DE–deltoideus pars acromialis; BB–biceps brachii; FC–flexor carpi ulnaris).

**Table 2. The results of the Wilcoxon Sign Test showing differences in the musculoskeletal load (RMS) of the upper limbs between the first and the sixth session of the bimanual coordination training.**

|  | % | Z | p |
|---|---|---|---|
| FC | 45.1 | 3.29 | 0.001 |
| DE | 52.4 | 1.57 | 0.116 |
| BI | 28.8 | 14.26 | < 0.001 |
| TR | 44.0 | 4.00 | < 0.001 |

with age [34], and the activation of the corpus callosum is associated with the performance of continuous coordination tasks [35]. It was also considered that the activation of the frontal cortex during shape-following bimanual coordination tasks results in increased functionality of sensorimotor connections [36,37]. Therefore, the tasks required visual information in the form of a fixed shape, or additionally in the form of a moving cursor.

The results of the tests performance indicate that the quality of control improved over the training sessions. In the sixth training session, the error indictors value were about half of the values recorded in the first session. The increase in control skills was also confirmed by the results of the 2HAND and DL tests. Also studies by Voelcker-Rehage et al. [33] indicate that the elderly may significantly improve their functions through coordination training. Some of studies examined the effect of retention interval in one week [21,24,25,27], one month [27], eight weeks [21] and six months [28]. All those studies showed that performance measures kept their values after the retention interval regardless the type of coordination task that was under study. In present study there were six trainings sessions implemented, which is more than in other studies. Study of Hoff et al. [15] in which retention was calculated as the difference in the amount of measure between successive training sessions showed lack of differences in retention between young and old people. Therefore, based on the previous results it was assumed that the retention effect will be sustained and it was not tested. To conduct long-period observation would gain more knowledge on the subject. However, it seems it would not

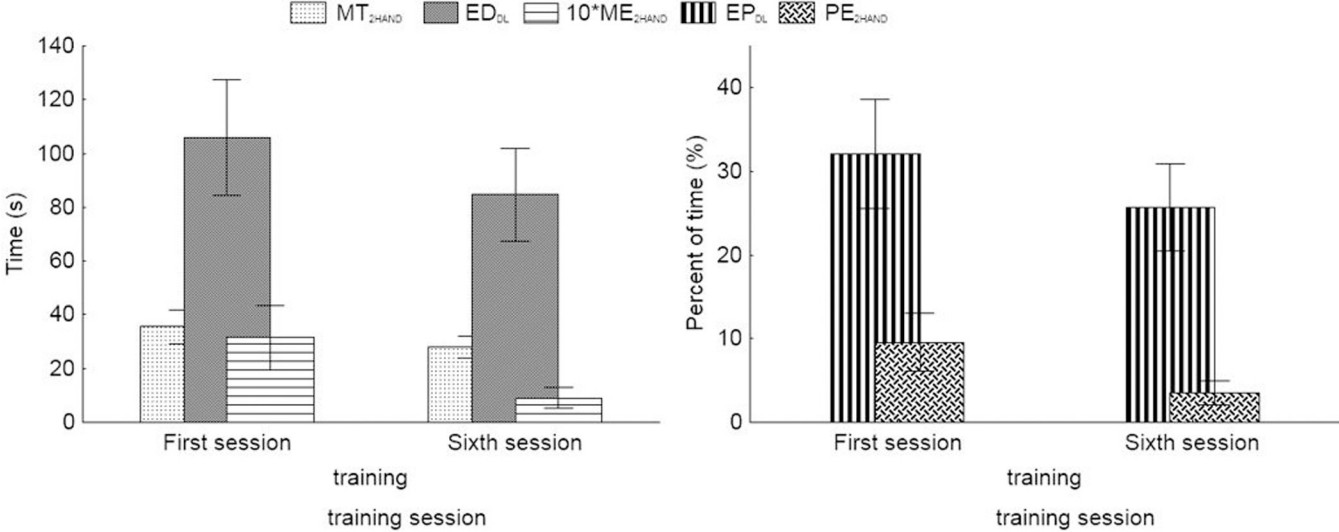

**Fig 7. Mean values and 95% CI of the results of the tests of the Vienna test measures.** 2HAND (MT$_{2HAND}$—Total duration; ME$_{2HAND}$—Total error duration; PE$_{2HAND}$—Total percent error duration) and Double Labyrinth (DL) (ED$_{DL}$—Error duration; EP$_{DL}$—Duration of the error in relation to the duration of the test) tests carried out during the first and during the sixth training session.

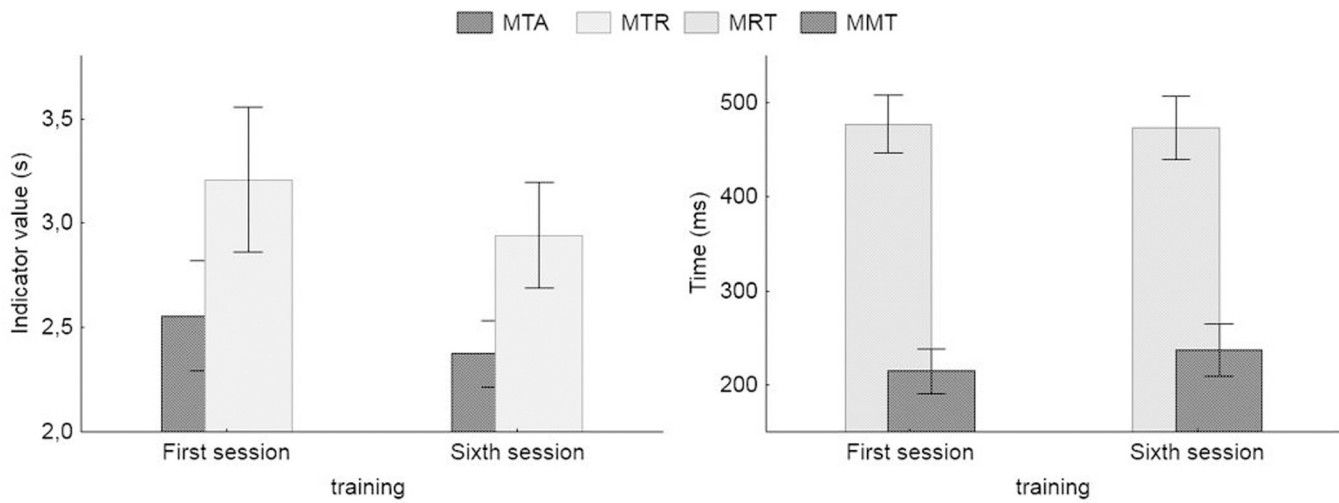

**Fig 8. Mean values and 95% CI of the results of the Vienna test measures.** COG (MTR—Time of correctly rejected; MTA—Time of correctly accepted) and of the RT (MRT—Reaction time; MMT—Motor time) tests carried out during the first and during the sixth training session.

provide different results compared to previous studies. Then it can be assumed that performing the developed bimanual coordination training has a positive effect on developing coordination skills, and this training can be a useful tool in mitigating age-related changes. However, progress in improvement of motor skills depends on the type of task, the feedback provided, and the duration of practice [33, 38].

In this study, the results obtained from the training depend on the type of tasks performed. The smallest changes in the error indicators occurred for FP and IP tasks, that is for symmetric tasks performed in a phase. Symmetric coordination standards require each of the two limbs to perform the same activity so that one limb performs a mirror reflection of the other. Global research has mainly focused on the difference in the performance of in phase and anti-phase tasks. The results of those tasks performance clearly show that bimanual coordination tasks performed in the anti-phase require increased precision from elderly people [39]. Also complex coordination tasks during which cooperation between limbs occurs at various amplitudes of movement or forces, directions and frequencies (rate of change), require higher levels of motor skills [40, 41]. The greatest changes in error indicators, i.e. the greatest potential for improving motor functions, occurred in the ICOa task and then in the IAP task. The ICOa task was a complex task, in which the movement of one cursor consisted of changes in the values of force of both upper limbs. Both tasks were tasks with the cursor movement speed imposed, and the tasks differed in coordination mode. Therefore, it may be assumed that coordination mode is important for improving coordination skills, and that tasks in the anti-phase and complex tasks are recommended for this purpose.

Tracking mode also plays a role in the values of error indicators for the tasks performed. In tasks with the imposed cursor movement speed, the decrease in values of the error indicators were higher than those for tasks with a freely chosen speed. It should also be noted that for the IP task, which is a task in the phase with the imposed speed, there is a particularly large difference between the decline of Error and the decline of Variability. This is also the only task for which the decline in Error was greater than the decline in Variability. In the IP task, the speed was imposed, and the shape of the cursor movement was more complicated than in other tasks. This may imply that the complexity of the shape has a greater impact on the enhancement of skills in the Error area rather than in Variability.

**Table 3.  The results of the Wilcoxon Sign Test showing differences in the values of the Vienna test measures between the first and the sixth session of the bimanual coordination training.**

|  | % | Z | p |
|---|---|---|---|
| $MT_{2HAND}$ | 34.61 | 1.373 | 0.170 |
| $ME_{2HAND}$ | 0.00 | 4.903 | < 0.001 |
| $PE_{2HAND}$ | 7.69 | 4.118 | < 0.001 |
| $ED_{DL}$ | 19.23 | 2.942 | 0.003 |
| $EP_{DL}$ | 19.23 | 2.942 | 0.003 |
| MTA | 36.00 | 1.200 | 0.230 |
| MTR | 32.00 | 1.600 | 0.110 |
| MMT | 73.08 | 2.157 | 0.031 |
| MRT | 38.46 | 0.981 | 0.327 |

The performance of manipulative activities with high accuracy increases the level of muscular activity [42,43]. In the present study, improvement in control skills was accompanied by a decrease in the muscular activity of the upper limbs. Normalized amplitude values demonstrate that from among the 4 muscles, only the DE muscle lacks the statistically significant differences indicating a decrease in load during the sixth training session compared to the first session. This may imply that greater control skills cause lower musculoskeletal load and it can be inferred that the activation of upper limb muscles improved. It means that the increase of functionality in terms of bimanual coordination resulted in a reduction of the musculoskeletal loading associated with the performance of activities requiring high coordination and precision. Such results suggest, that appropriate training can help to mitigate the effects of degradation of motor processes linked to ageing. However, especially in the case of the TR muscle, the reduction of the load may be associated with a decline in mental load, which was caused by the performance of known tasks [44].

Learning is thought to rely on the ability of neuronal network to modify their structure and function with specific stimuli or environmental exposure, that is, neuroplasticity [45]. The development of tasks included in the training presented in these studies used the premise of the impact on brain regions through bimanual coordination performance. It was expected that due to the fact that bimanual coordination tasks are linked with a more complex modulation of connections within and between the hemispheres [13], this would lead to the activation of brain areas impacting cognitive functions. Activation of additional brain areas positively correlates with the complexity of performed tasks [46]. It was demonstrated that coordination training may lead to the activation of the parts of the brain responsible for cognitive functions, such as attention [47]. However, in the study presented, the results of the COG test showed no statistically significant differences between the first and the last training session, and therefore did not confirm the impact of the tasks covered by bimanual coordination on the cognitive functions of information processing and attention. By introducing coordination training, Johann et al. [10] obtained similar results, i.e. an increase of motor coordination in the absence of an impact on cognitive functions. Therefore, it can be assumed that the impact of bimanual coordination training on cognitive functions is not explicit and may be more dependent on the characteristics of individual subjects. A series of biological, psychological, health, environmental and lifestyle factors may be significant here [48]. The intelligence quotient, education, type of work performed, and participating in intellectually stimulating activities are associated with maintaining cognitive functions at a higher level [49]. That could have impact on alleged changes in cognitive functions in group of the present study participants.

## Limitations of the study

The analysis carried out and its results have some limitations. Although each of tasks characteristic features were standardized as much as possible, individual variations and comparably small sample size may have contributed to the lack of significant differences in the outcome variables of cognitive measures. Another important aspect is the fact that the effectiveness of training can depend on the combined impact of several personal characteristics, such as e.g. gender, obesity, diseases or motor lateralization, which may affect bimanual coordination in different ways. To the best of our knowledge, there is no research that would provide information on the effects of gender or obesity on learning bimanual coordination. However, even if gender is an important factor in the ability to bimanual coordination learning, it does not affect the present study, as the study group had the same number of women and men. It was demonstrated that limb domination affects bimanual coordination, since people with restricted domination show better results [50], and lateral specialization has a negative effect on the performance of symmetric tasks [51]. Also the results might be biased by diseases, especially of neurodegenerative origin, which means that the study conclusions can refer only to healthy, older people.

To be effective, training that improves the functionality of elderly workers should produce significant, measurable and validated results. This means that it should involve specific activities for the functioning of elderly workers and which can be used both for rehabilitation purposes and provide indicators of neurodegenerative diseases, e.g. to identify clinical symptoms of Parkinson's disease [52,53]. An analysis that takes into account the different combinations of bimanual coordination features simultaneously would give a broad assessment of the impact of different characteristics on the suitability of training for improving the quality of coordination. The developed tasks, that were included in the training were characterized by only three features, which narrows the assessment of the tasks of bimanual coordination.

## Conclusion

Results of present studies that evaluated the impact of training involving different bimanual coordination tasks on the precision of motor activities of the elderly, healthy population proved that the performance of bimanual coordination tasks has a positive effect on the improvement of coordination skills. It can be also concluded that:

- the tasks included in the training although had a positive effect on motor functions did not affect cognitive functions such as attention and information processing;

- coordination mode is important for improving coordination skills, and tasks in the anti-phase and complex tasks are recommended for this purpose;

- tracking mode plays a significant role, tasks with an imposed cursor movement speed have greater potential to improve coordination skills than tasks with freely chosen speed;

- improved control skills result in the reduction of upper limb muscular load, it can therefore be assumed that an increase in control skills with the use of appropriate training can help reduce the musculoskeletal load when performing work activities requiring precision and coordination.

## Supporting information

**S1 File. Dataset.**
(XLSX)

## Acknowledgments

This paper is based on the results of a research task carried out within the scope of the fourth stage of the "Improvement of safety and working conditions" National Programme in 2017–2019—within the scope of research and development—by the Ministry of Science and Higher Education/National Centre for Research and Development. The Central Institute for Labour Protection–National Research Institute (CIOP-PIB) is the Programme's main coordinator.

## Author Contributions

**Conceptualization:** Danuta Roman-Liu, Zofia Mockałło.

**Data curation:** Danuta Roman-Liu.

**Formal analysis:** Zofia Mockałło.

**Investigation:** Zofia Mockałło.

**Methodology:** Danuta Roman-Liu, Zofia Mockałło.

**Supervision:** Danuta Roman-Liu.

**Writing – review & editing:** Danuta Roman-Liu.

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
