## [Decision Letter · Decision Letter 0]

28 Nov 2019

PONE-D-19-30238

Effectiveness of bimanual coordination tasks performance in improving coordination skills and cognitive functions in elderly

PLOS ONE

Dear Prof. Roman-Liu,

Thank you for submitting your manuscript to PLOS ONE. After careful consideration, we feel that it has merit but does not fully meet PLOS ONE’s publication criteria as it currently stands. Therefore, we invite you to submit a revised version of the manuscript that addresses the points raised during the review process.

We would appreciate receiving your revised manuscript by Jan 12 2020 11:59PM. To enhance the reproducibility of your results, we recommend that if applicable you deposit your laboratory protocols in protocols.io, where a protocol can be assigned its own identifier (DOI) such that it can be cited independently in the future. For instructions see: http://journals.plos.org/plosone/s/submission-guidelines#loc-laboratory-protocols

We look forward to receiving your revised manuscript.

Kind regards,

Feng Chen

Academic Editor

PLOS ONE

Journal Requirements:

1. We note that you have stated that you will provide repository information for your data at acceptance. Should your manuscript be accepted for publication, we will hold it until you provide the relevant accession numbers or DOIs necessary to access your data. If you wish to make changes to your Data Availability statement, please describe these changes in your cover letter and we will update your Data Availability statement to reflect the information you provide.

2. Please remove your figures from within your manuscript file, leaving only the individual TIFF/EPS image files, uploaded separately.  These will be automatically included in the reviewers’ PDF.

Reviewers' comments:

Reviewer's Responses to Questions

**Comments to the Author**

1. Is the manuscript technically sound, and do the data support the conclusions?

Reviewer #1: Partly

Reviewer #2: Yes

2. Has the statistical analysis been performed appropriately and rigorously? 

Reviewer #1: N/A

Reviewer #2: Yes

3. Have the authors made all data underlying the findings in their manuscript fully available?

Reviewer #1: Yes

Reviewer #2: Yes

4. Is the manuscript presented in an intelligible fashion and written in standard English?

Reviewer #1: No

Reviewer #2: Yes

5. Review Comments to the Author

Reviewer #1: 1. The Participants‘s attributes are varied, which would affect the research conclusions.

2. The study completed successive training during 2-3 days aims to the same group, it’s common sense to get some positive effects. However, What about the duration of effects? Is it necessary to conduct long-period observation?

3. More previous studies should be addressed in the introduction part. The gap between previous studies and this research should be clearly described.

4. The manuscript is hard to read, cause the fonts and layout are inappropriate.

Reviewer #2: aThe topic of this paper is important and the methods sound. The results are meaningful. There are several suggestions to improve this paper.

1. For “51 and 58%”, it will be better to be changed to “51-58%” or “51% and 58%”.

2. For “mean reaction time”, the definition of reaction time is needed, which the authors could refer to the following paper. And does the “mean” means the average reaction time of all participants? It is not clear.

[1] “Examining the safety of trucks under crosswind at bridge-tunnel section: A driving simulator study”, Tunnelling and Underground Space Technology, 2019, 92, 103034.

3. There should be some references for Wilcoxon test.

4. In the conclusion part, the author could list some major findings in this paper.

6. PLOS authors have the option to publish the peer review history of their article (what does this mean?). If published, this will include your full peer review and any attached files.

Reviewer #1: No

Reviewer #2: No

---

## [Decision Letter · Decision Letter 1]

21 Jan 2020

Effectiveness of bimanual coordination tasks performance in improving coordination skills and cognitive functions in elderly

PONE-D-19-30238R1

Dear Dr. Roman-Liu,

We are pleased to inform you that your manuscript has been judged scientifically suitable for publication and will be formally accepted for publication once it complies with all outstanding technical requirements.

With kind regards,

Feng Chen

Academic Editor

PLOS ONE

Additional Editor Comments (optional):

Reviewers' comments:

Reviewer's Responses to Questions

**Comments to the Author**

1. If the authors have adequately addressed your comments raised in a previous round of review and you feel that this manuscript is now acceptable for publication, you may indicate that here to bypass the “Comments to the Author” section, enter your conflict of interest statement in the “Confidential to Editor” section, and submit your "Accept" recommendation.

Reviewer #1: All comments have been addressed

Reviewer #2: All comments have been addressed

2. Is the manuscript technically sound, and do the data support the conclusions?

Reviewer #1: Yes

Reviewer #2: Yes

3. Has the statistical analysis been performed appropriately and rigorously? 

Reviewer #1: Yes

Reviewer #2: Yes

4. Have the authors made all data underlying the findings in their manuscript fully available?

Reviewer #1: Yes

Reviewer #2: Yes

5. Is the manuscript presented in an intelligible fashion and written in standard English?

Reviewer #1: Yes

Reviewer #2: Yes

6. Review Comments to the Author

Reviewer #1: (No Response)

Reviewer #2: (No Response)

7. PLOS authors have the option to publish the peer review history of their article (what does this mean?). If published, this will include your full peer review and any attached files.

Reviewer #1: No

Reviewer #2: No

---

## [Editor Report · Acceptance letter]

24 Feb 2020

PONE-D-19-30238R1 

Effectiveness of bimanual coordination tasks performance in improving coordination skills and cognitive functions in elderly 

Dear Dr. Roman-Liu:

I am pleased to inform you that your manuscript has been deemed suitable for publication in PLOS ONE. Congratulations! Your manuscript is now with our production department. 

With kind regards,

on behalf of

Dr. Feng Chen 

Academic Editor

PLOS ONE